

# Recognising depression in non-human primates: a narrative review of reported signs of depression

Jonas C. P. van Oosten[1], Annemie Ploeger[2] and Elisabeth H. M. Sterck[1,3]

[1] Animal Behaviour and Cognition, Department of Biology, Utrecht University, Utrecht, Netherlands
[2] Developmental Psychology, Department of Psychology, University of Amsterdam, Amsterdam, Netherlands
[3] Animal Science Department, Biomedical Primate Research Centre, Rijswijk, Netherlands

## ABSTRACT

Major depressive disorder (depression) is a highly heterogenous human mental disorder that may have equivalents in non-human animals. Research into non-human depression teaches us about human depression and can contribute to enhance welfare of non-human animals. Here, we narratively review how signs of depression in non-human primates (NHPs) can be observed based on symptoms of the Diagnostic and Statistical Manual of Mental Disorders (DSM-5). Furthermore, we propose diagnostic criteria of NHP depression and we review reports on signs of depression in NHPs. We diagnose an NHP with depression when it shows a core sign (depressed mood or anhedonia) alongside at least three other DSM-5-derived signs of depression. Results show that four out of six observable signs of depression are present in NHPs, occasionally lasting for months. However, only a group of six NHPs in one study met our proposed criteria for a diagnosis of depression. We call for more research into the co-occurrence of depressive symptoms in individual NHPs to establish the prevalence of depression in NHPs.

## INTRODUCTION

In humans, major depressive disorder, or simply depression, is a highly heterogenous mental disorder. Depression may have equivalents in other species, including our closest living relatives, non-human primates (NHPs). Accordingly, NHPs display behaviours that may indicate depression (*e.g.*, *Ausderau et al., 2023*; *Ferdowsian et al., 2011*; *Harlow & Suomi, 1974*; *Hinde, Spencer-Booth & Bruce, 1966*; *MacLellan et al., 2021*; *Shively et al., 1997*). In humans, depression is estimated to affect approximately 185 million people globally, constituting 2.5% of the human population (*Institute of Health Metrics and Evaluation, 2019*). According to the fifth edition of the Diagnostic and Statistical Manual of Mental Disorders (DSM-5; *American Psychiatric Association, 2013*), symptoms of depression include a depressed mood, anhedonia, differences in weight or appetite, insomnia or hypersomnia, psychomotor agitation or retardation, fatigue or loss of energy, feelings of worthlessness or excessive or inappropriate guilt, diminished ability to think or

Corresponding author
Jonas C. P. van Oosten,
jonas@vanoosten.nl

concentrate or indecisiveness, and thoughts of death or suicide (left column of Table 1). For depression to be diagnosed in humans, the DSM-5 requires the co-occurrence of at least five symptoms in the same 2-week period, at least one being a "core symptom" (*i.e.*, depressed mood or anhedonia). In humans, symptoms are typically assessed by verbal reports of patients, sometimes supplemented by observations (*American Psychiatric Association, 2013*). Non-human animals cannot speak, so, to study depression in NHPs, each human depressive symptom must be translated to an observable sign for NHPs.

It is important to first address the question whether NHPs can experience depression at all. Multiple lines of reasoning show support for a homology between depression in humans and other animals (*MacLellan et al., 2021*). First, homology is widely assumed in the large variety of animal models of depression (*Becker, Pinhasov & Ornoy, 2021*; *Gururajan et al., 2019*; *Petković & Chaudhury, 2022*) and primates (*Bliss-Moreau & Rudebeck, 2021*). In animal models of depression, animals are artificially brought into a state in which they display at least one sign of depression, although individuals differ in their reactions (*Anisman & Matheson, 2005*). Furthermore, affect is present across mammalian species (*e.g.*, *Lagisz et al., 2020*; *Mendl et al., 2009*) is considered homologous across mammals (*Bliss-Moreau & Rudebeck, 2021*). Additionally, it is suggested that negative (and positive) affective states are evolutionarily conserved (*Mason, 2010*). Negative states are adaptive when the animal is confronted with a stressor, promoting behaviours to mitigate the stressor (*Denver, 2009*; *Mason, 2010*). This homology in the mechanisms of negative affect suggests homology of psychopathology, including depression, in mammalian species. Altogether, these lines of reasoning support the notion that NHPs can experience depression.

Depression research in non-humans benefits both non-humans and humans. Studying depression in NHPs as model species is relevant because of the high evolutionary proximity between humans and other primates (*e.g.*, *Bliss-Moreau & Rudebeck, 2021*; *Czéh & Simon, 2021*). Furthermore, understanding depression in non-human species may contribute to understanding the evolution of depression, which may improve treatment in humans (*Durisko, Mulsant & Andrews, 2015*; *Nettle & Bateson, 2012*). Additionally, studying behaviour resembling depression can aid in improving animal welfare (*e.g.*, *Lecorps, Weary & von Keyserlingk, 2021*; *Úbeda et al., 2021*). Diagnoses of depression open the door for interventions for captive animals, such as improving housing facilities (*Lecorps, Weary & von Keyserlingk, 2021*; *MacLellan et al., 2021*), or administration of antidepressants (*Chu, 2019*). Furthermore, the mental state of animals can influence their behaviour, so monitoring animals' mental state can benefit the validity of animal studies (*Camus et al., 2015*). Moreover, differentiating behavioural patterns (*e.g.*, depression and anxiety) might result in a better understanding of the relation between the environment and behaviours, which allows for more specific interventions. Ultimately, humans have an ethical responsibility to care for captive animals, so it is valuable to know whether an NHP is depressed.

The suggested continuity of mammalian mechanisms responsible for depression suggests that depression occurs in NHPs. The aim of this narrative review is to assess whether NHPs experience depression. We propose diagnostic criteria of NHP depression

**Table 1** An overview of the DSM-5 diagnostic criteria of major depressive disorder (depression) in humans and corresponding behavioural phenotypes in non-human primates.

| Human diagnostic criterion (DSM-5) | Observable behaviours in NHPs (our proposal) |
| --- | --- |
| (A) Presence of five or more of the following symptoms; at least one being depressed mood or anhedonia. | |
| S1: Depressed mood | Pessimistic judgement in judgement bias task (*Lagisz et al., 2020*) |
| S2: Diminished interest or pleasure (anhedonia) | Diminished social and sexual behaviour (*Paredes, 2009*, *2014*; *Trezza, Campolongo & Vanderschuren, 2011*; *Vanderschuren, Achterberg & Trezza, 2016*) |
| | No increase in anticipatory behaviour when expecting reward (*Von Frijtag et al., 2000*) |
| S3: Differences in weight or appetite | Differences in food intake (*Zijlmans et al., 2021*) |
| | Changes in weight (≥5%)/morphology indicating weight change (*Berman & Schwartz, 1988*; *Zijlmans, Langermans & Sterck, 2019*) |
| S4: Insomnia or hypersomnia | Circadian rhythm (24-h observations or focal observations) (*Hennessy, Chun & Capitanio, 2017*) |
| | Sleep fragmentation (*Ayuso et al., 2023*) |
| | Nocturnal activity (*Green, 2018*; *Kooros et al., 2022*; *Schork et al., 2023*, *2024*) |
| S5: Psychomotor agitation (S5a) or retardation (S5r) | Higher (agitation) or lower (retardation) frequency and/or duration of locomotion (*Li et al., 2020*; *Qin et al., 2015*; *Teng et al., 2021*) |
| | Hunched posture (retardation) (*e.g.*, *Kaufman & Rosenblum, 1967*; *Shively et al., 1997*) |
| S6: Fatigue or loss of energy | *Not observable in NHPs* |
| S7: Feelings of worthlessness or excessive or inappropriate guilt | *Not observable in NHPs* |
| S8: Diminished ability to think[1] or concentrate, indecisiveness | Diminished short-term or working memory in a delayed matching-to-sample task (*e.g.*, *Truppa et al., 2014*) |
| S9: Recurrent suicidal thoughts, ideation, plans or attempts | *Not applicable in NHPs* |
| (B) The symptoms cause significant distress. | |
| (C) The episode is not attributable to effects of a substance or another medical condition. | |
| (D) The symptoms are not better explained by schizophrenia spectrum or other psychotic disorders. | |
| (E) There has never been a manic episode or a hypomanic episode. | |

**Note:**
[1] We use the term "processing information" rather than thinking, as it has not been decisively established whether non-human animals are able to think (*e.g.*, *Phillips et al., 2021*; but see *Kano & Call, 2021*).

based on the DSM-5, supplemented by primate specific behaviour. Accordingly, we review NHP literature of reported signs of depression, attributable to one change in the environment (*e.g.*, a sudden change in husbandry, but not arrays of multiple stressors). We then use our proposed diagnostic criteria of NHP depression to assess whether NHPs are depressed. This review is intended for an audience of researchers interested in depression and welfare in nonhuman primates, and for those interested in the evolutionary background of depression. We outline below how we identified several signs of depression in NHPs in the literature and identified only one study where NHPs would be diagnosed with depression according to our criteria.

## RECOGNISING DEPRESSION IN NON-HUMAN PRIMATES

In this review, we will assess whether and how the nine human symptoms under criterion A of the DSM-5 diagnostic criteria of major depressive disorder (Table 1) can be recognised in non-human primates. There have been previous efforts to apply human diagnostic criteria of depression, such as the DSM-5 or the International Classification of

Diseases (*World Health Organization, 2019*) to non-humans (*Ferdowsian et al., 2011*; *MacLellan et al., 2021*; *Úbeda et al., 2021*). In one study, criteria of the DSM-IV (*American Psychiatric Association, 2000*) were applied to behavioural phenotypes in chimpanzees (*Pan troglodytes*; *Ferdowsian et al., 2011*). Depressed mood, anhedonia, eating/weight changes, diurnal rhythm changes, psychomotor changes, and cognitive deficits were considered observable, yet a limitation is their use of subjective assessments. Instead, we argue that objectively observable behaviours should be used to assess signs of depression (*Rosati et al., 2012*).

We take the human DSM-5 as a guideline for assessing signs of depression in NHPs and supplement it with NHP specific behaviour, basing the diagnostic criteria of NHP depression on observable behaviours that NHPs show. The reason we chose the DSM-5 as a starting point is not because we anthropomorphise NHPs, but because most depression research is done in humans and the disorder has only been confirmed in humans. As the DSM-5 diagnostic criteria of depression form a validated framework for human diagnosis, we used the DSM-5's structure to assess depression in non-humans. Still, we are open to add non-DSM-5-derived behaviours to our diagnostic criteria, if these behaviours are clearly associated with depression.

Table 1 shows the DSM-5 criteria A–E for depression. We did not use criteria B–E, because significant distress (criterion B) is not observable and other (mental) illnesses are not within the scope of this review (criteria C–E). We will not include cases where medication is used (criterion C).

We aimed to develop a diagnostic approach that is usable across many different contexts, ranging from (semi-)wild to captive, and that is minimally intrusive for the animals and feasible for researchers. Therefore, we used observational methods over tasks that require training, with the latter requiring time and effort from researchers, and not being feasible in wild settings. Additionally, when an NHP is suspected to be depressed, it may not cooperate in training for tasks due to anhedonia or cognitive deficits (*Horne, Topp & Quigley, 2021*; *Wooldridge et al., 2021*). When there is no valid observational method available, we recommend tasks that require minimal to no training. Only when both these approaches are unavailable or invalid, we discuss tasks that require substantial training.

In the following subsections, we discuss whether the nine DSM-5 symptoms of depression can be measured through observation in NHPs (Table 1).

## Depressed mood (S1)

Depressed mood (*i.e.*, being sad, hopeless or discouraged) is one of the two core symptoms of depression (*American Psychiatric Association, 2013*), being reported by 97% of patients with depression (*Bryan et al., 2008*). If not self-reported, depressed mood can be assessed by reading the person's facial expression and demeanour, for example, by observing tears (*American Psychiatric Association, 2013*). However, in NHPs, there is no evidence that certain facial expressions or demeanors indicate a depressed mood (*Kret et al., 2020*). Still, two behaviours accompanying a depressed mood can be observed in both humans and NHPs: judgement biases and attentional biases.

Depressed humans are more likely to interpret neutral cues, such as neutral facial expressions, as negative (*Everaert, Podina & Koster, 2017*; *Bourke, Douglas & Porter, 2010*, *Maniglio et al., 2014*; *Van Vleet et al., 2019*). These biases can be assessed by judgement bias tests (JBTs; *Lagisz et al., 2020*; *Mendl et al., 2009*). In JBTs it is assumed that judgement and attention are influenced by emotional state (*Coleman & Pierre, 2014*). JBTs have been successfully used in a variety of vertebrates (*Bateson & Nettle, 2015*; *Douglas et al., 2012*; *McCoy et al., 2019*) and even in insects (*Baracchi, Lihoreau & Giurfa, 2017*). Non-human animals can be trained to recognise positive and negative cues, and their level of optimism or pessimism can be measured subsequently (*Mendl et al., 2009*). For example, JBTs have been used to assess the judgements of rhesus macaques (*Macaca mulatta*) as a function of husbandry conditions (*Bethell et al., 2012a*). Although the overall effect sizes of results from JBTs are small, manipulations generally had the expected effect, validating JBTs as an indicator of affect in non-human animals (*Lagisz et al., 2020*). A downside of JBTs is that they require substantial training (*e.g.*, *Bateson & Nettle, 2015*). Therefore, while JBTs validly measure depressed mood, applying the task is not feasible.

Another cognitive bias in depressed humans is an increased attention towards negative cues, such as sad faces (*Keller et al., 2019*; *Peckham, McHugh & Otto, 2010*). Several tasks with varying complexity measure attentional biases in NHPs (*Bethell & Pfefferle, 2023*; *Crump, Arnott & Bethell, 2018*): preferential looking time tasks, dot-probe tasks, and emotional Stroop tasks.

In a preferential looking task (*Winters, Dubuc & Higham, 2015*), an NHP watches a side by side on-screen presentation of a neutral and a negative stimulus (*e.g.*, a neutral and an aggressive facial expression, *Bethell et al., 2012b*). Depressed humans tend to look more at the negative stimulus after a stressful event. Interestingly, rhesus macaques showed the opposite pattern after a veterinary check (*Bethell et al., 2012b*), possibly indicating avoidance of threatening behaviour. Moreover, although stressful, a veterinary check is not expected to induce depression. In contrast, sheep (*Ovis aries*) that were pharmacologically treated to be in a depression-like state did prefer to look towards a threatening compared to a positive stimulus (*Monk et al., 2018*). Because of the contrasting evidence, the preferential looking task is not (yet) valid to study depressed mood in NHPs.

Depressed humans also show attentional biases in dot-probe tasks (*Peckham, McHugh & Otto, 2010*), in which a neutral and negative stimulus are displayed side by side on a screen. When the images disappear, one stimulus is replaced by the probe, typically consisting of two dots. Humans are instructed to respond to this dot-probe as quickly as possible (*e.g.*, by touching it on the screen). The latency to respond the dot-probe is lower when the subject's attention was on the stimulus that was replaced by the dot-probe, thereby revealing biases in attention (*Van Rooijen, Ploeger & Kret, 2017*). Dot-probe tasks were successfully adapted for rhesus monkeys (*Parr et al., 2013*), bonobos (*Pan paniscus*: *Kret et al., 2016*), chimpanzees (*Wilson & Tomonaga, 2018*), and long-tailed macaques (cynomolgus monkeys, *Macaca fascicularis*; *Cassidy et al., 2023*). NHPs need to be familiarised with the set-up and trained to respond to the dot-probe. In bonobos, this required a 15-week training programme (*Kret et al., 2016*). However, more work is needed to validate whether increased attention to a specific stimulus is a sign of depression in

NHPs. Furthermore, human performance in dot-probe tasks is also influenced by anxiety (*Van Rooijen, Ploeger & Kret, 2017*). This makes it difficult to disentangle effects of the two disorders. In short, dot-probe tasks might elicit changes in attention resulting from depression, but because of validity issues and the requirement of extensive training, this task is not very suitable for measuring depressed mood in NHPs.

A third way of measuring attentional biases is with an emotional Stroop task (*Stroop, 1935*), in which depressed and healthy humans perform differently (*Joyal et al., 2019*; *Peckham, McHugh & Otto, 2010*). The emotional Stroop task is based on the notion that emotion distracts an individual from performing a task, such as naming the colour that a word is written in *Crump, Arnott & Bethell (2018)*. Participants are instructed to name this colour as quickly and accurately as possible. When the content of the words is negative (*e.g.*, pain), the response time increases. The emotional Stroop task has been adapted for multiple NHP species (*Allritz, Call & Borkenau, 2016*; *Hopper et al., 2021*; *Laméris et al., 2022*). In short, NHPs are trained to choose a picture over another adjacent picture based on differently coloured picture borders on a touch screen. The content of the picture can reflect positive, neutral, or negative information. The reaction time is then compared between the three content categories (*Allritz, Call & Borkenau, 2016*). Despite the adaptation for NHPs, no studies have yet used the task as an indicator of depressed mood. A major limitation of the emotional Stroop task is the inability to rule out that the differences in reaction time reflect biases in motor actions (*Crump, Arnott & Bethell, 2018*). Moreover, training NHPs to participate in the emotional Stroop task can take up to thousands of trials, if successful (*Allritz, Call & Borkenau, 2016*; *Hopper et al., 2021*). For these reasons, we do not consider the emotional Stroop task a valid or feasible measure of depressed mood in NHPs.

Altogether, attentional bias tasks are not valid measures for depressed mood in NHPs. Judgment bias tasks are suitable, although these are not easily implementable because training is required.

## Anhedonia (S2)

The DSM-5 states that a patient suffering from anhedonia, reported in 71% of depressed humans (*Cao et al., 2019*), does not find pleasure in previously rewarding activities (*American Psychiatric Association, 2013*). There is a distinction between the experience of rewarding stimuli (consummatory anhedonia) and the motivation to pursue rewarding stimuli (motivational anhedonia) (*Treadway & Zald, 2011*).

Anhedonia is observable, as it may result in social withdrawal and reduction of sexual interest (*American Psychiatric Association, 2013*). Behaviours like social play, social interaction with offspring, and sexual behaviour are rewarding for both humans and other animals (*Paredes, 2009*, *2014*; *Trezza, Campolongo & Vanderschuren, 2011*; *Vanderschuren, Achterberg & Trezza, 2016*) and can be observed with a relatively low labour investment. As the systems and mechanisms of sociality are homologous across mammals (*Crockford et al., 2013*; *Massen, Sterck & de Vos, 2010*), motivational anhedonia may result in a decrease in social and sexual behaviours in NHPs and humans alike. Still, social and sexual behaviours can change when the group composition changes

(*Caselli et al., 2023*; *Ryan & Hauber, 2016*; *Schel et al., 2013*). However, if this alternative explanation is ruled out, diminished social behaviours can be a valid indicator of motivational anhedonia in NHPs.

Sugar consumption is another rewarding behaviour in humans and other animals (*Hajnal, Smith & Norgren, 2004*; *Olszewski et al., 2019*). Therefore, reduced preference of sweetened water (sucrose) over unsweetened water may indicate decreased reward sensitivity and thus anhedonia in NHPs (*Li et al., 2020*; *Qin et al., 2015*; *Teng et al., 2021*), yet this measure of anhedonia is not undisputed. Sucrose preference decreased in maternally deprived rhesus monkeys compared to a non-deprived control group, but the monkeys also consumed more bitter (quinine) water (*Paul, English & Halaris, 2000*). Thus, the monkeys did not display anhedonia, but rather became more insensitive to flavours (*Paul, English & Halaris, 2000*). Moreover, the translational value of sucrose preference in non-humans to anhedonia in humans is still unclear (*Markov, 2022*). In addition, setting up a sucrose preference test involves a labour investment. Altogether, more research is needed to validate sucrose preference as a measure of anhedonia in NHPs.

Furthermore, humans suffering from depression show less anticipation towards rewards (*Rzepa, Fisk & McCabe, 2017*; *Stringaris et al., 2018*). Anticipatory behaviour is defined as a change in behaviour prior to an anticipated rewarding event and can also be measured in non-humans: when animals expect a reward, they present either increased or decreased activity (*Krebs et al., 2017*; *Von Frijtag et al., 2000*). Anticipation of rewards represents an appetitive behaviour (*Von Frijtag et al., 2000*), indicative of reward sensitivity (*Kamal et al., 2010*; *Van der Harst & Spruijt, 2007*; *Watters, 2014*). Although anticipatory behaviour is considered a welfare indicator in various mammalian species (*Buchanan-Smith, 2011*, *Dudink et al., 2006*, *Vinke, Van Den & Spruijt, 2004*), it has not yet been used as an indicator of anhedonia in NHPs. As an anhedonic NHP experiences less pleasure from rewards, it may also have a decreased reward sensitivity (*Wooldridge et al., 2021*). In this way, anticipatory behaviour can be used to assess anhedonia in NHPs. Because captive animals often learn to associate feeding moments with other stimuli *via* classical conditioning (*e.g.*, *Dickinson & Mackintosh, 1978*), anticipatory behaviour can be easily observed in captivity. Thus, although anticipatory behaviour has not yet been used to assess anhedonia in NHPs, we highlight its potential as an indicator.

Overall, social behaviour is the most valid and most feasible way of assessing anhedonia in NHPs. We do not currently regard diminished sucrose preference as evidence of anhedonia and propose anticipatory behaviour as a new measure of anhedonia in NHPs.

## Differences in appetite or weight (S3)

According to the DSM-5, depression may decrease or increase appetite, which can impact food intake and consequently, lead to weight change (*American Psychiatric Association, 2013*). A total of 40% of depressed patients report weight gain, while 30% report weight loss (*Caroleo et al., 2019*). More than 5% weight change in a month is considered significant in the DSM-5. In NHPs, appetite changes cannot be directly observed, but food intake can be observed without disturbing the animal. Animals can also be weighed, although that

requires either capturing the animal or training it to voluntarily cooperate in a weighing procedure (*Zijlmans et al., 2021*). Alternatively, body condition scoring can be used as a semi-quantitative method to reliably assess body fat and muscle (*Clingerman & Summers, 2005*). However, this requires the animal to be immobilised, which is highly intrusive. Instead, obesity class can be assessed reliably by observations using the Obesity scale (*Berman & Schwartz, 1988*). This allows for the determination of obvious weight change without the need to disturb the animal. Similar to the DSM-5, a change in obesity class refers to a weight difference of ≥5% (*Zijlmans, Langermans & Sterck, 2019*). Motivation for food could also be used as a measure for appetite, but as an animal that is not motivated for food may also be anhedonic, motivation for food is not a truly valid measure for appetite. In short, two valid and feasible ways to assess appetite and weight changes in NHPs exist: observing an animal's food intake and visually observing weight change. Voluntary weighing is a feasible option for trained individuals.

### Insomnia or hypersomnia (S4)

Depression is associated with both insomnia and hypersomnia. Insomnia is estimated to be present in 50–90% of depressed patients, whereas 16–20% of patients report hypersomnia (*Tsuno & Ritchie, 2005*). In NHPs, sleep patterns can be observed. Observing an animal's full circadian rhythm provides the most valid indication of sleep patterns, yet is very labour intensive. Alternatively, a valid indication of sleep patterns may be drawn by using systematic short focal samples (*Altmann, 1974*; *Hennessy, Chun & Capitanio, 2017*). Moreover, sleep data may be acquired through automated methods (*Rushen, Chapinal & de Passilé, 2012*; *Whitham & Miller, 2016*; *Green, 2018*). For example, non-invasive audio recording devices capture sounds that indicate sleep interruptions (*Ayuso et al., 2023*). However, these devices have only been used to measure sleep at the group level. Alternatively, individual activity can be measured with accelerometers in captive and wild NHPs (*Kooros et al., 2022*; *Schork et al., 2023*). Activity patterns during the night indicate periods of sleeping and waking. However, putting on collars requires handling the animals. Lastly, sleep patterns can be derived from automated methods to measure movement from (infra-red) videos (*Green, 2018*; *Schork et al., 2024*). While still not well developed to measure individual sleep and requiring visibility of the sleeping individual, this forms an interesting future avenue to measure sleep non-invasively.

In sum, several methods exist for assessing sleep disturbances in NHPs. These concern, at a group level, nocturnal audio monitoring and automated infra-red recording. At the individual level, non-invasive systematic focal sampling can be used, whereas employing accelerometers is a more invasive method.

### Psychomotor agitation or retardation (S5)

The DSM-5 specifically states that psychomotor agitation or retardation must be visible by others, rendering this an observable symptom in humans (*American Psychiatric Association, 2013*). Psychomotor agitation is present in 10–46% of depressed patients (*Sobin & Sackeim, 1997*) and its characteristics (difficulties sitting still, pacing, or various ways of fidgeting) can be observed or assessed through self-report (*American Psychiatric Association, 2013*). In

NHPs, psychomotor retardation is sometimes assessed using open field tests (*e.g.*, *Li et al., 2020*; *Teng et al., 2021*). More or less locomotion in open field tests may reflect psychomotor changes (*Grønli et al., 2005*), However, open field tests were not designed to assess psychomotor changes, so their validity is disputed (*MacLellan et al., 2021*). Alternatively, psychomotor changes may be better reflected by activity in the home environment (*MacLellan et al., 2021*), which can be measured non-invasively by locomotion frequency and duration (*e.g.*, *Li et al., 2020*; *Teng et al., 2021*). Thus, instead of using the disputed open field test, psychomotor agitation is best assessed in NHPs by measuring activity in the animal's home environment.

The DSM-5 characterises psychomotor retardation with slowed speech, thinking, and body movements, which have to be observable by others (*American Psychiatric Association, 2013*). Humans with psychomotor retardation exhibit a fixed gaze, slower movement, decreased movement, and a slumped posture (*Buyukdura, McClintock & Croarkin, 2011*). A total of 62% of patients with depression report psychomotor retardation (*Bryan et al., 2008*). However, psychomotor retardation can also be a residual effect of past depressive episodes in humans (*Gorwood et al., 2014*). In NHPs, reduced locomotion frequency and duration is often used as an indicator for psychomotor retardation, either in the home environment or open field (*e.g.*, *Li et al., 2020*; *Teng et al., 2021*). As mentioned in the previous paragraph, locomotion changes in the animal's home environment are a feasible and valid measure of psychomotor retardation, but we cautiously treat the outcomes of open field tests.

Furthermore, the slumped body posture observed in humans with psychomotor retardation shows similarities with a behaviour in NHPs that is called hunching (*e.g.*, *Kaufman & Rosenblum, 1967*; *Lopresti-Goodman, Kameka & Dube, 2012*; *Shively et al., 1997*). Hunching is defined as a "slumped or collapsed body posture", while having open eyes to distinguish from sleeping (*Shively et al., 1997*). Hunching is recorded in pigtail macaques (*Macaca nemestrina*; *Kaufman & Rosenblum, 1967*), bonnet macaques (*Macaca radiata*; *Rosenblum & Paully, 1984*), rhesus macaques (*Camus et al., 2014*; *Hinde, Spencer-Booth & Bruce, 1966*), long-tailed macaques (*Shively et al., 1997*), and in chimpanzees (*Goodall, 1986*; *Lopresti-Goodman, Kameka & Dube, 2012*). Because of the great similarity with human posture changes, hunching is a good observable indicator of psychomotor retardation in NHPs.

Altogether, psychomotor agitation and retardation can be assessed non-invasively by recording locomotion frequency and duration. Additionally, hunching can be used as a non-invasive indicator for psychomotor retardation.

## Fatigue or loss of energy (S6)

A depressed human patient may report fatigue without physical exertion and find that simple tasks require more time and energy (*American Psychiatric Association, 2013*). A total of 90% of depressed humans report feelings of fatigue or loss of energy (*Ghanean, Ceniti & Kennedy, 2018*). As fatigue is a subjective experience, it is impossible to observe it directly in humans. However, reported fatigue is associated with a decrease in physical activity in humans, although the direction of causality remains unclear (*Puetz, 2006*).

NHPs likely can experience fatigue, because there is homology in the mechanisms responsible for affect across primates (*Bliss-Moreau & Rudebeck, 2021*; *Lagisz et al., 2020*). Accordingly, decreased locomotion is typically used as an indication of fatigue in NHPs (*Li et al., 2020*; *Teng et al., 2021*). However, locomotion is also used to assess psychomotor retardation. So, it is not possible to distinguish fatigue from psychomotor retardation. Similarly, fatigue is associated with lower levels of sleep (*e.g.*, *Darwent et al., 2015*), but this is already another criterion of depression (see "Differences in appetite or weight (S3)"). Thus, although NHPs may experience fatigue, no measures to independently assess fatigue currently exist.

## Feelings of worthlessness or guilt (S7)

The DSM-5 names feelings of worthlessness or excessive or inappropriate guilt as a symptom of depression, disclosed by the patient through self-report (*American Psychiatric Association, 2013*). Feelings of guilt are more prominent in humans with depression than controls (*Luck & Luck-Sikorski, 2021*). Furthermore, humans with depression have feelings of exaggerated responsibility for uncontrollable events (*Kim, Thibodeau & Jorgensen, 2011*). In human children and adults, feelings of guilt have been linked to observations of increased reparative prosocial behaviours, such as apologising or stating concern (*Bybee, Merisca & Velasco, 1998*; *Donohue & Tully, 2019*). Subsequently, it has been suggested that prosocial approach behaviours can be used as an indicator of feelings of guilt in non-human species (*Malhotra, 2019*). However, social behaviours can also be diminished as a result of anhedonia (see "Anhedonia (S2)"). Furthermore, the DSM-5 states that the guilt must be excessive or inappropriate (*American Psychiatric Association, 2013*), so just assessing guilt would not be sufficient. The next step requires to examine when guilt is excessive, which is currently impossible in non-verbal animals. Altogether, feelings of worthlessness/guilt currently cannot be recognised in NHPs.

## Diminished ability to think or concentrate, indecisiveness (S8)

The DSM-5 states that an impaired ability to think, concentrate, or make decisions (referred to as cognitive deficits) is often present in depression (*American Psychiatric Association, 2013*). This is often assessed by self-report, but distraction or impaired "short-term or working" memory can be observed (*American Psychiatric Association, 2013*; *de Almeida et al., 2012*; *Dillon & Pizzagalli, 2018*; *Douglas et al., 2009*; *Marazziti et al., 2010*). The DSM-5 uses the term "think" in this criterion. However, it has not been decisively established whether non-human animals are able to think, *i.e.*, reflect on their behaviour, situation or emotions (*e.g.*, *Phillips et al., 2021*; but see *Kano & Call, 2021*). Because of this, we will use the term "processing information" instead of "thinking" for NHPs.

A measure of short-term and working memory is the delayed matching-to-sample task (DMTS), in which depressed humans make more errors (*Douglas et al., 2009*; *Porter et al., 2003*; *Shah et al., 1999*), and which can also be used in NHPs (*e.g.*, *Chen et al., 2023*; *Truppa et al., 2014*). In a DMTS, a stimulus is presented to the subject. After a short interval, the initial stimulus is presented along with a novel stimulus. The subject is instructed (for humans) or rewarded (for NHPs) to select the initial stimulus (*Chudasama, 2010*).

Although the DMTS has never been used to study depression in NHPs, the DMTS could be a valid indicator of cognitive deficits. A downside of the DMTS is the need for training the participating animals, requiring an extensive time and labour investment. Additionally, because the DMTS measures both short-term and working memory, effects on either of those are impossible to distinguish, Thus, although a valid task to measure cognitive deficits, the use of the DMTS has some drawbacks.

An alternative short-term memory task that requires no training and only comprises two trials is a preferential looking task (see "Depressed mood (S1)") with two neutral stimuli (*Ennaceur & de Souza Silva, 2018*), sometimes referred to as a recognition memory task (*Capitanio, 2021*). In the first trial, subjects are familiarised with a stimulus. In the second trial, the familiar stimulus is presented next to a novel stimulus. A healthy individual spends more time looking at the novel stimuli, whereas NHPs with adverse experiences do not spend more time looking at the novel stimuli (*Capitanio, 2021*), similar to a paradigm to study recognition memory in human infants (*Fantz, 1964*; *Rose, Feldman & Jankowski, 2004*). In humans, results are conflicting: some researchers find that depression negatively impacts recognition memory of previously shown words (*Deijen, Orlebeke & Rijsdijk, 1993*; *Watts, 2014*), whereas others do not find a relationship (*Brand, Jolles & Gispen-de Wied, 1992*; *Dunbar & Lishman, 1984*). In sum, due to conflicting results in humans, it is unclear whether recognition memory tasks in NHPs validly represent cognitive deficits associated with depression.

In summary, cognitive deficits are measurable in NHPs, but not by mere observation. The delayed matching-to-sample task is a valid measure of impaired cognitive function. The preferential looking task with neutral stimuli might be more feasible, but validity is currently lacking.

## Suicidal thoughts, ideation, plans or attempts (S9)

Humans with depression may think about death more often, may be wishing not to be alive, or may even plan and attempt suicide (*American Psychiatric Association, 2013*). Suicidal ideation is typically elicited through interviewing patients and is present in 11–63% of the depressed population (*Vuorilehto et al., 2014*). For suicidal behaviour to occur, an understanding of death is required. NHPs have a basic understanding of the concept of death of others, but it remains unclear whether NHPs have a causal understanding of events leading to death (*Gonçalves & Carvalho, 2019*). This means we cannot assume that NHPs have "thoughts" about death or suicide, and that we should not attribute self-harming or self-endangering behaviours (*Walker et al., 2012*) to suicidal ideations in NHPs. In sum, there is no measure to assess whether NHPs are capable of having suicidal ideations, so suicidal ideation cannot be used as a criterion of depression in NHPs.

## Conclusion: proposed diagnostic criteria of NHP depression

In short, we consider six DSM-5 signs of depression observable in NHPs: depressed mood, anhedonia, appetite/weight changes, diurnal rhythm changes, psychomotor changes, and cognitive deficits. Three DSM-5 signs are not observable: fatigue/loss of energy, feelings of

worthlessness/guilt, and suicidality. Therefore, directly applying the human diagnostic criteria for diagnosing depression (at least one core symptom alongside four other symptoms; *American Psychiatric Association, 2013*) to non-humans, would likely result in a large number of false negative diagnoses. To prevent this, we propose less "strict" diagnostic criteria for non-human depression, compared to the human criteria. We propose the following diagnostic criteria for depression in non-human primates:

> *A non-human primate is considered depressed when it shows at least one core sign of depression (depressed mood or anhedonia), alongside at least three other signs (appetite/weight changes, diurnal rhythm changes, psychomotor changes, cognitive deficits) during the same two-week period.*

In the next part of this review, we will apply the proposed diagnostic criteria to cases of potentially depressed NHPs.

## REPORTS OF NON-HUMAN PRIMATE DEPRESSION

In this section, we narratively review studies where a single change in the environment was associated with one or more signs of depression. Although we acknowledge that depression can occur from an accumulation of stressors, we chose this scope because the cause of depression can be more easily identified when focussing on single events. The first (JvO) and the second (AP) author used the PubMed, Google Scholar, and Web of Science electronic databases for studies published until October 1st, 2024. The following search terms were used: (non-human primate OR nonhuman primate OR monkey OR ape) AND (depression OR depressive symptoms). JvO and AP also scanned the reference lists of relevant articles and checked articles citing relevant work. English, peer-reviewed journal articles of original studies were screened by JvO. From the 4,945 results, 14 studies were included that reported signs of depression in nonhuman primates attributable to one change in the environment. Four additional studies were found in reference and "cited-by" lists. We added two examples from books and one example based on personal communication. This added up to 21 sources (Table 2). Although we have tried to cover as much literature on depression in nonhuman primates as possible, we like to stress that this is a narrative review, with the aim to unravel whether cases of depression in NHPs are available in the literature, following a list of diagnostic criteria.

We distinguish between experimentally induced and naturally occurring (non-experimentally induced) signs. For each case, we compare the observed signs to our proposed diagnostic criteria, thereby identifying potential risk factors of depression in NHPs. If an individual showed depressed mood or anhedonia alongside at least three other signs of depression for at least 2 weeks, it is diagnosed with depression. A summary of the results is displayed in Table 2.

### Experimental induction of signs of depression

In the 1960s and 1970s, induction of depression in NHPs became a topic of interest (*Harlow & Suomi, 1974*). Signs of depression were induced in young rhesus macaques by separating them from their mothers and through use of social isolation and social deprivation. In addition, manipulation of social hierarchies (*Shively et al., 1997*) and the

**Table 2 An overview of studies in which one environmental change was associated with one or more signs of depression.** For each study, we provide the species, number of subjects (N), rearing history, housing type, the distinction between experimentally induced and naturally occurring depression, the risk factor, the level of analysis (group or individual), and the level of evidence (experimental, correlational, descriptive, or anecdotal). Furthermore, we show whether our criteria for NHP depression were met: the presence of at least one core sign (depressed mood or anhedonia) alongside at least three other signs during the same two-week period. We classified a sign as present when it was observed using a valid indicator (see "Recognising depression in nonhuman primates"). When reports are unclear or alternative explanations cannot be ruled out (see "Reports of non-human primate depression"), we classified the sign as "maybe present", marked with a question mark. We refer to the six observable signs of depression as follows: S1 = depressed mood, S2 = anhedonia, S3 = diurnal rhythm changes, S4 = eating/weight differences, S5a = psychomotor agitation, S5r = psychomotor retardation, S8 = cognitive deficits. Fatigue/energy loss, (S6), worthlessness/guilt (S7) and suicidality (S9) cannot be observed in NHPs. One group of six individuals met our diagnostic criteria for NHP depression (Boccia et al., 1997).

| Subjects | N | Rearing history | Housing | Context | Risk factor | Diagnostic criteria for NHP depression | | | Source | Group/ individual level | Level of evidence |
|---|---|---|---|---|---|---|---|---|---|---|---|
| | | | | | | ≥1 core sign (s present) | ≥3 other signs (s present) | ≥2 weeks | | | |
| Juvenile *Callithrix jacchus* (7–9 m old) | 15 (7♀) | Social | Group, in/outdoor not stated | Experimental | Social isolation | No | No (S3) | Yes | da Silva et al. (2018) | Group | Experimental |
| ♂ adult *Mandrillus sphinx* | 4 | Wild | Semifree-ranging | Natural | Status loss | Maybe (S2?) | No (S3) | Not clear | Setchell & Dixson (2001) | Group | Correlational |
| ♂ adult *Papio hamadryas* | 1 | Wild | Wild | Natural | Status loss | No | No (S3, S5r?) | Not clear | Kummer (1995) | Individual | Anecdotal |
| ♀ adult *Macaca fascicularis* | 42 | Not stated | Quarantine → group, in/outdoor not stated | Experimental | Status loss | No | No (S3, S5r) | Yes | Shively et al. (1997) | Group | Experimental |
| ♀ adult *M. fascicularis* | 5 (+20 controls) | Social | Indoor with daylight | Experimental | 90 d social isolation | Maybe (S2?) | No (S5r) | Not measured | Li et al. (2013) | Group | Experimental |
| ♀ adult *M. fascicularis* | 5 (+20 controls) | Social | Indoor with daylight | Experimental | 90 d social and visual isolation | Maybe (S2?) | No (S5r) | Not measured | Li et al. (2013) | Group | Experimental |
| ♀♂ adult *M. fascicularis* and *M. mulatta* | Unknown | Social | Group, in+outdoor | Natural | Status loss | Yes (S2) | No (S3, S5r) | Mostly not | A Louwerse, 2023, personal communication | Individual | Anecdotal |
| ♂ infant *M. mulatta* (45 d old) | 4 | Individual | Pre: individual; post: group, in/outdoor not stated | Experimental | Vertical chamber confinement | Maybe (S2?) | No (S5r?) | Yes | Suomi & Harlow (1972) | Group | Experimental |
| ♂ infant *M. mulatta* (11 w old) | 1 | Wild | Wild | Natural | Maternal loss | Yes (S2) | No (S5r) | Yes | Berman (1982) | Individual | Correlational/ descriptive |
| Infant *M. mulatta* (16–32 w old) | 23 (11♀) | Wild | Wild | Natural | More time away from mother | For 3 indiv.: yes (S2) | For 3 indiv.: no (S5r) | Not stated, probably yes | Berman, Rasmussen & Suomi (1994) | Individual | Descriptive |
| Infant *M. mulatta* (21–32 w old) | 22? (not clearly reported) | Not stated, probably mother-reared | Group | Experimental | Maternal separation | Yes (S2) | No (S5r) | No | Hinde & Spencer-Booth (1971) | Group | Descriptive |
| Infant *M. mulatta* (30–32 w old) | 4 | Mother-reared | Group in+outdoor | Experimental | Maternal separation | Yes (S2) | No (S5r) | No | Hinde, Spencer-Booth & Bruce (1966) | Group | Descriptive |
| Infant *M. mulatta* | 13 (+6–8 controls) | Social | Group, in+outdoor | Experimental | Maternal separation | No | No (S5) | Yes | Spencer-Booth & Hinde (1971) | Group | Experimental |
| ♂ adolescent *M. mulatta* (3 y old) | 4 (+4 controls) | Social (except 3w separation) | Individual, indoor | Experimental | Vertical chamber confinement | Maybe (S2?) | No (S5r?) | Yes | McKinney, Suomi & Harlow (1972) | Group | Experimental |
| ♂ *M. mulatta* (4–6 y old) | 12 (+12 controls) | Mother-reared | Group, outdoor | Experimental | Individual indoor housing | No | No (S4, S5r) | No | Hennessy, Chun & Capitanio (2017) | Group | Experimental |
| ♀ adult *M. mulatta* | 8 | Not stated | Individual, indoor | Experimental | Shorter photoperiod | No | No (S3, S5r) | Yes | Qin et al. (2015) | Group | Experimental |
| ♂ adult *M. mulatta* | 20 (+20 controls) | Social | Group | Experimental | Individual housing | No | No (S5r) | Yes | Jackson et al. (2023) | Group | Experimental |

(Continued)

| Subjects | N | Rearing history | Housing | Context | Risk factor | Diagnostic criteria for NHP depression | | | | Group/ individual level | Level of evidence |
| | | | | | | ≥1 core sign (s present) | ≥3 other signs (s present) | ≥2 weeks | Source | | |
|---|---|---|---|---|---|---|---|---|---|---|---|
| Infant *M. nemestrina* (5–6 m old) | 4 | Social | Group, in/outdoor not stated | Experimental | Maternal separation | Yes (S2) | No (S5r) | Yes | Kaufman & Rosenblum (1967) | Group | Experimental |
| Infant *M. radiata* | 5 (3♀) | Mother-reared | Group, indoor | Experimental | Maternal separation | No | No ( S5r) | No | Reite, Kaemingk & Boccia (1989) | Group | Experimental |
| *M. radiata* (4–6 m old) | 6 | Social | Group, indoor | Experimental | Maternal separation | Yes (S2) | Yes (S3, S4, S5r) | Yes | Boccia et al. (1997) | Group | Experimental |
| ♀ infant *Gorilla gorilla gorilla* (2 y 8 m old) | 1 | Social | In+ outdoor | Experimental | Maternal separation | Yes (S2) | No (none) | No | Nakamichi, Silldorff & Sexton (2001) | Individual | Correlational |
| *Pan troglodytes* (14 m–9 y old) | 13 (8♀) | Wild | Wild | Natural | Maternal loss | Maybe (S2?) | No (S3? S5r) | Yes | Goodall (1986) | Individual | Anecdotal |

photoperiod (*Qin et al., 2015*) led to signs of depression. In this section, we explore whether animals subjected to these experiments show depression following our criteria.

### Maternal separation

One of the first practices to induce signs of depression was the separation of infant rhesus, bonnet and pigtail macaques from their mothers (*Worlein, 2014*). In these experiments, mothers were removed from the social group and the behaviour of their infants was recorded. Rhesus monkeys who were separated from their mother show a consistent behavioural pattern (*Harlow & Suomi, 1974*), which was recognised as the protest and despair stages in maternally separated human children (*Bowlby, 1960*). The macaque protest stage lasts a few days and is characterised by increased communication calls, crying, and locomotion (*Seay, Hansen & Harlow, 1962*). Although increased locomotion represents a sign of depression, the increased locomotion in the protest stage lasts less than 2 weeks and likely results from searching behaviour. In the despair stage, the infants typically showed hunching, slow movements, and self-clutching (*Harlow & Suomi, 1974*).

In five studies on maternal separation (details in Table 2), infants showed psychomotor retardation after separation (*Hinde, Spencer-Booth & Bruce, 1966*; *Hinde & Spencer-Booth, 1971*; *Kaufman & Rosenblum, 1967*; *Reite, Kaemingk & Boccia, 1989*; *Spencer-Booth & Hinde, 1971*). A reduction in social play (anhedonia) was observed in two studies (*Hinde, Spencer-Booth & Bruce, 1966*; *Kaufman & Rosenblum, 1967*). Sleep was only assessed in one study, and no differences were found (*Reite, Kaemingk & Boccia, 1989*). Most signs of depression faded after reunification with the mother, but some rhesus macaques showed psychomotor retardation up to 6 months after reunification (*Spencer-Booth & Hinde, 1971*).

Although no maternally separated individual in the above mentioned studies could be diagnosed with depression, one group of six bonnet macaque infants did meet our criteria. Infants and their mothers were removed from the social group and separated, before the infant was let back into the group for a 2-week separation (*Boccia et al., 1997*). Compared to the pre-separation baseline, the infants showed more eating, whereas sleep, social play, and locomotion decreased (*Boccia et al., 1997*). Because these subjects showed core sign anhedonia alongside increased food intake, insomnia and psychomotor retardation over a 2-week separation, these six infants meet our criteria for NHP depression.

Behaviour associated with maternal separation was also studied opportunistically in a zoo-housed infant western lowland gorilla (*Gorilla gorilla gorilla*; *Nakamichi, Silldorff & Sexton, 2001*). The mother of the 2.5-year-old female infant was removed from the social group for 9 days for medical reasons. During this period, the infant showed one sign of depression: diminished social play. However, anhedonia alone is not sufficient to meet the proposed diagnostic criteria.

In short, several maternal separation studies showed no conclusive evidence for depression, but one group of six bonnet macaque infants met our criteria for diagnosis. These subjects showed one core sign of depression (anhedonia) alongside three additional signs (increased appetite, insomnia, psychomotor retardation).

### Vertical chamber confinement

The vertical chamber is an apparatus that was designed to induce depression in rhesus macaques, described as "a stainless-steel chamber, open at the top with sides that slope downward and inward to form a rounded bottom" (*Harlow & Suomi, 1974*). Immature monkeys were placed in the vertical chamber for weeks, with no space to move or access to conspecifics. Four 45-day old individually reared male rhesus monkey infants were confined in the vertical chamber for 45 days. After release, "chambered" monkeys showed more self-clasping, more hunching and less locomotion, less environmental exploration, and virtually no social behaviours compared to control groups (statistically significant, *Suomi & Harlow, 1972*). These behaviours remained at stable levels in the 8 months following release from the vertical chamber (*Suomi & Harlow, 1972*). However, the near absence of social behaviour may also be attributable to the lack of socialisation of these individually reared subjects. Additionally, the decreased locomotion may also be explained by physical pain or impaired development caused by the confinement. Although the infants might have shown two signs of depression (anhedonia, psychomotor retardation) for more than 2 weeks, no individual met the criteria for a depression diagnosis.

In another study, four 3-year old rhesus males were subjected to 10 weeks in the vertical chamber. The macaques had been socially reared (except a 3-week separation period) and had been housed individually for a year before confinement in the vertical chamber. After release back into group housing, the monkeys were observed for 3 weeks. Statistical analysis between groups showed that prolonged clinging increased and locomotion decreased compared to before confinement and to an undisturbed control group (*McKinney, Suomi & Harlow, 1972*). Again, decreased locomotion may also be explained by physical pain resulting from the confinement. Other than clinging, social interactions and play were diminished. Although these are indicators of anhedonia, the observed increase in contact clinging seemingly contradicts this. Therefore, we are unsure whether anhedonia was present in this group. Unlike the chambered infants described above (*Suomi & Harlow, 1972*), the adolescent monkeys did not show hunching (*McKinney, Suomi & Harlow, 1972*). Because the chambered adolescent rhesus macaques may only have shown two signs of depression (anhedonia, psychomotor retardation), no individual met our criteria for NHP depression.

### Hierarchy manipulation

In an experiment (*Shively et al., 1997*) several novel social groups of four female long-tailed macaques, a highly despotic species, were formed ($N = 42$). In these groups the monkeys formed linear hierarchies. The monkeys at positions one and two in the hierarchies were classified as dominant, and the other two as subordinate. Next, groups were reshuffled so that the new groups consisted of either dominant or subordinate individuals. Previously dominant individuals could become subordinate and previously subordinate individuals could become dominant, or individuals maintained a similar hierarchy position. After the reshuffling, behaviour was observed for 26 months and compared between groups. During this period, prevalence of hunching was highest in subordinate-subordinate monkeys (83%) and lowest in dominant-dominant individuals (0%). Dominant-subordinate

monkeys (36%) and subordinate-dominant monkeys (25%) were intermediate. As the prevalence of hunching before reshuffling was not reported, it is unclear whether hunching is a result of rank loss or of subordination in general. Subordinates spent significantly more time hunching compared to current dominants and had less interest in food (*Shively et al., 1997*). Despite the presence of two signs of depression (decreased appetite, psychomotor retardation) persisting for well over 2 weeks, this was not sufficient to meet our proposed diagnostic criteria of depression.

### Social isolation

More recently, the effects of changing from social to solitary housing were explored. Macaques in outdoor, large-group housing were rehoused in two conditions: individual indoor housing ($N = 12$) or indoor housing with an affiliative partner ($N = 12$). Individuals spent 8 days in the experimental condition (wave 1), then 2 weeks back in their original group, and then another 8 days in the same experimental condition (wave 2). During the two waves, behaviour was observed and analysed between groups. In wave 1, the individually housed and pair-housed groups did not statistically differ in durations of hunching or activity. However, individually housed subjects spent significantly less time sleeping compared to pair-housed macaques. During wave 2, individually housed subjects spent significantly more time hunching and less time active, yet showed no significant differences in sleep time compared to pair-housed individuals (*Hennessy, Chun & Capitanio, 2017*). Altogether, individually rehoused rhesus macaques show at most two signs of depression for less than 2 weeks, so no individual met the criteria for a diagnosis.

A similar study examined the behavioural effects of within-institution rehousing on group-living adult male rhesus monkeys (*Jackson et al., 2023*). To adjust to their transfer, for 4 weeks twenty monkeys acclimatised in pairs, whereas twenty others acclimatised in individual housing. The singly-housed group showed significantly more hunching (55% prevalence) than the pair-housed group (0% prevalence) (*Jackson et al., 2023*). Although the psychomotor retardation persisted throughout the 4-week acclimatisation period, showing only one symptom of depression does not meet our criteria for diagnosis.

In another study, healthy adult female long-tailed macaques were randomly distributed over three groups: social isolation ($N = 5$), social and visual isolation ($N = 5$) and group-housed controls ($N = 20$) (*Li et al., 2013*). Researchers observed both before and after the 90-day isolation and the data were analysed on the group level. Comparison of pre-isolation and post-isolation data showed no difference in weight in both isolated groups. After being reunited with their social group, the monkeys showed less embracing and lip-smacking (social behaviours), but no difference in (received) grooming compared to pre-separation. One sexual behaviour (presenting) decreased after social isolation, but the frequency of same-sex mounting and copulations did not change. Because the results on the different social and sexual behaviours were inconsistent, the presence of anhedonia cannot be confirmed. For the group social and visual isolation group, less alloparental behaviour and less sexual presenting and same-sex mounting (but not copulation) was observed post-isolation compared to pre-isolation (*Li et al., 2013*). The decrease in two sexual behaviours suggests anhedonia. Both isolated groups showed more hunching

behaviour, which indicates psychomotor retardation (*Li et al., 2013*). In short, socially and visually separated long-tailed macaques exhibit no more than two co-occurring signs of depression, the duration of which was unclear. Thus, criteria for diagnosis were not met.

In another study, fifteen juvenile common marmosets (*Callithrix jacchus*) of equal sex distribution were moved to individual housing for 8 weeks. Behaviour was observed at before moving, in the first and last week of isolation (*da Silva et al., 2018*). Group-level comparisons between the baseline and isolation period yielded no difference in locomotion. Nevertheless, feeding time and body weight significantly decreased. Additionally, more "somnolence" (sleepiness, possibly representing insomnia) was observed (*da Silva et al., 2018*). In short, the authors aimed to measure three signs of depression, but only one sign (appetite/weight changes) could be observed with certainty, thereby not meeting the diagnostic criteria for depression.

Altogether, social isolation could lead to three signs of depression (appetite/weight change, insomnia, psychomotor retardation), yet none of the NHPs could be diagnosed with depression.

### Manipulation of photoperiod

To model seasonal affective disorder, the light-dark cycle of eight female rhesus macaques was manipulated from a 12 h:12 h light:dark ratio to a 5 h:19 h ratio (*Qin et al., 2015*; comparable to mid-winter at 62° latitude, *Time and Date, 2024*). After 3 months of the 5 h:19 h ratio, group-level statistical analysis showed that the monkeys weighed significantly less, showed more hunching, and showed less locomotion compared to pre-intervention. Despite showing two signs of depression (weight change, psychomotor retardation), no individual rhesus monkey can be diagnosed with depression.

### Do NHPs show depression in experimental contexts?

Altogether, the different experimental procedures led to several different signs of depression. Only in one out of fifteen studies, a group of NHPs met our diagnostic criteria for depression. The seemingly low prevalence of depression in NHPs possibly results from studies failing to assess the co-occurrence of all signs of depression. Thereby, these cases are rendered incomplete in terms of the evaluation of diagnostic criteria. Another important limitation is that the reported experimental studies measure signs of depression on a group level instead of an individual level, which does not provide enough detail to assess signs of depression in individual NHPs.

## Naturally occurring signs of depression

Besides experimental contexts, signs of depression can also occur naturally. Here, we discuss signs of depression that are associated with maternal loss, natural social separation, and social status loss.

### Maternal loss

Maternal loss is associated with signs of depression in dependent offspring in chimpanzees and rhesus macaques. Thirteen young chimpanzees at Gombe (ages between 14 months and 9 years) lost their mothers in an anecdotal report (*Goodall, 1986*). A description of

every individual's reaction was provided. From these thirteen, three pre-weaning age orphans did not survive without their mother's milk. Four post-weaning individuals (4–9 years) showed none to very few behavioural differences after maternal death, maybe because of less dependency on their mothers. However, up to six post-weaning individuals (3–8.5 years) did show signs of depression (*Goodall, 1986*). The frequency of play in these individuals dropped (probably social play included, suggesting anhedonia) after maternal death. Three of the six individuals died after showing some signs of depression, yet the cause of death could not be established. One infant showed hunching behaviour and another individual's tool using skills deteriorated, which might represent a cognitive deficit. In two individuals, a potbelly was observed, which likely is a hunger oedema caused by starvation, possibly indicating depression-induced weight loss. An alternative explanation could be less access to resources after maternal death. One individual was reported to show no sexual interest until age 6 or 7 years, possibly indicating anhedonia. Alternatively, this could also be attributed to the individual's slower physical sexual development (*Goodall, 1986*). Despite the signs persisting for more than 2 weeks in all six individuals, none of the individuals met the criteria to be diagnosed with depression.

In a wild rhesus macaque population, an 11-week old male infant lost his mother (*Berman, 1982*). From the day following the mother's disappearance, researchers anecdotally reported hunching and more inactivity. Additionally, the infant seemed less responsive to initiations of social contact by group members. Descriptive quantitative data were also provided, comparing the orphaned infant to 20 unorphaned infants from the same group. The number of play initiations by the orphaned infant dropped from around the median before the mother's disappearance to almost zero after. The infant also rejected play initiations more often than any other infant (*Berman, 1982*). All these behaviours gradually regressed to normal over the time span of 16 weeks. Altogether, this infant likely showed psychomotor retardation and anhedonia following maternal loss, although statistical evidence is absent. Either way, this individual did not meet the diagnostic criteria for depression.

Altogether, whereas maternal loss may lead to four signs is depression (anhedonia, psychomotor retardation, and possibly weight change, cognitive deficits), no individual NHP could be diagnosed with depression.

### Natural social separation

Similarly to maternal loss, natural changes in the mother-infant relationship are sometimes associated with signs of depression in infant rhesus macaques. In the same wild population as described in the previous case, researchers compared the mother-infant relationship and infant behaviours before and after the start of the mating season (*Berman, Rasmussen & Suomi, 1994*). A total of 23 infants (11 females) were observed before and after their mothers resumed mating. After the resumption of mating, the infants spent more time out of sight of their mothers or out of physical contact with their mothers. This situation shows parallels to acute separation due to death or experimental intervention, although mothers and infants still had contact everyday (*Berman, Rasmussen & Suomi, 1994*). On the group level, no signs of depression were identified, but three infants showed

hunching and six infants showed a decrease in play after their mothers resumed mating. These descriptive results do not state if any infant simultaneously showed two signs of depression (psychomotor retardation, anhedonia), but even then, these infants would not meet the criteria for diagnosis.

### Status loss

Loss of high social status (a top or high position in the dominance hierarchy) is associated with signs of depression in NHPs. Anecdotal evidence of signs of depression was found in wild male hamadryas baboons (*Papio hamadryas*) after status loss, *i.e.*, a drop from the highest position in the dominance hierarchy to a lower position (*Kummer, 1995*). While not omnipresent, one defeated male "visibly lost weight" and started walking "rather stiffly", yet this is not conclusive evidence of the presence of psychomotor retardation. So, this individual showed two signs of depression at most, which is not enough to diagnose depression.

Four semi-free-ranging mandrills (*Mandrillus sphinx*) lost statistically significant amounts of weight after loss of the alpha position (*Setchell & Dixson, 2001*). Alternatively, this may be explained by a reduced access to food resources. Additionally, one male spent significantly less time in the centre of the group after defeat (*Setchell & Dixson, 2001*). This social withdrawal could be a sign of anhedonia, but could also be a result of the changed group dynamics after the social defeat. Altogether, the defeated mandrill males did not display enough signs of depression to be diagnosed.

Signs of depression were also reported in socially defeated long-tailed and rhesus macaques in captive multigenerational groups. In males, sudden drops in dominance concern defeat of a male in the alpha position by another male in the group. In females, sudden drops in dominance are found during matriline overthrows, where lower ranking matrilines defeat the alpha family. Anecdotal evidence suggests that this status loss can lead to a specific behavioural phenotype in some of the males and females. These animals "sit in a corner in a hunched position, do not move, do not eat or drink, do not react to other group members and some show hypothermia. Despite veterinarian interventions, individuals that show these behaviours often die" (A. Louwerse, 2023, personal communication). Thus, defeated long-tailed and rhesus macaques show three signs of depression (psychomotor retardation, no eating, anhedonia). The signs often last until death, which is mostly within 2 weeks. This means that no individual defeated macaque met our diagnostic criteria for depression.

Altogether, status loss may result in three signs of depression (anhedonia, weight loss, psychomotor retardation).

### Does depression occur naturally in NHPs?

Although NHPs are reported to show signs of depression associated with maternal loss, natural social separation and status loss, no individual met our criteria for a diagnosis of NHP depression. Again, this may result from the lack of assessment of co-occurrence of signs of depression in these studies. Additionally, evidence for spontaneous occurrence of depression in NHPs is almost exclusively anecdotal, or occasionally correlational.

Although this level of evidence limits the strength of the conclusions that can be drawn, these studies are still valuable to inspire future research.

## DISCUSSION

In this narrative review, we clarified which signs of depression can be directly recognised by observation in NHPs and we set criteria to diagnose depression in NHPs that experienced one major change in their environment. We identified the presence of four to five signs of depression over all reviewed studies. Only in one study, a group of six maternally separated bonnet macaques met our diagnostic criteria for depression. Still, this does not mean that only one group of six NHPs suffered from depression. We argue that this lack of diagnoses is most likely explained by a failure to describe or assess the relevant signs. So, we do not regard our findings as evidence that NHPs cannot experience depression.

Additionally, we identified several environmental changes that may serve as risk factors for signs of depression, induced by experiments and occurring naturally, across different housing conditions and even in (semi-)wild NHPs. Potential risk factors of experimentally induced signs of depression included social deprivation, highly restricted control over the environment, and exposure to a shorter photoperiod. Naturally occurring signs of depression were only identified from correlational research or anecdotal reports, and were (seemingly) associated with maternal loss, social separation and status loss.

### Recognising signs of depression in NHPs

Four or five signs of depression were identified across the reviewed studies. Six out of nine DSM-5 symptoms of depression have corresponding observable signs in NHPs: depressed mood, anhedonia, differences in weight, differences in diurnal rhythm, changes in psychomotor speed, and cognitive deficits (see Table 1). Based on the observable signs, diagnostic criteria for NHP depression were formulated, requiring at least one of the core signs, depressed mood or anhedonia, to be present, alongside at least three other signs of depression during the same 2-week period. Based on the proposed criteria, we found experimental evidence for anhedonia, eating/weight changes, changes in sleep, and psychomotor speed changes. We found anecdotal evidence for cognitive deficits, and no evidence for the presence of depressed mood (see Table 2). In experimental contexts, anhedonia, eating/weight changes, changes in sleep, and psychomotor speed changes were observed. For naturally-occurring depression, only studies with lower levels of evidence were available. We found possible evidence for anhedonia, correlational evidence for weight changes and anecdotal evidence of psychomotor changes and cognitive deficits. Signs often persisted for longer than 2 weeks in both experimental and non-experimental contexts (*e.g.*, *Goodall, 1986*; *Qin et al., 2015*; *Shively et al., 1997*; *Spencer-Booth & Hinde, 1971*; *Suomi & Harlow, 1972*; see Table 2). However, only one group of NHPs in the reviewed studies (*Boccia et al., 1997*) met our diagnostic criteria for depression.

## Why were diagnoses of NHP depression so rare?

The question then arises why so few diagnoses of NHP depression were made in this review. First, we reiterate that the lack of diagnoses is not evidence of absence of depression in NHPs; we argue that the lack of diagnoses can mainly be explained by the lack of research into the co-occurrence of all depressive symptoms. Although six signs of depression can be observed in NHPs (as outlined above), no empirical study has thus far examined all six signs simultaneously. Previous research typically focussed only on easily observable signs, such as hunching behaviour, weight differences or changes in locomotion. In contrast, measuring depressed mood or cognitive deficits is more difficult, as this requires setting up a task. Indeed, no study explored these symptoms. We recommend future researchers to examine the co-occurrence of all measurable signs of depression during a period of 2 weeks or more. Considering that using suboptimal methods, some NHPs could still be diagnosed with depression (*Boccia et al., 1997*), we expect that more research into co-occurrence of the observable signs will result in more diagnoses of depression.

Another limitation contributing to the lack of diagnoses is that signs of depression are often measured and analysed on group level instead of on individual level. In most experimental studies, the aim is to find out whether certain circumstances can induce depression, which is analysed on the group level (*Harlow & Suomi, 1974*; *Hennessy, Chun & Capitanio, 2017*; *Qin et al., 2015*; *Shively et al., 1997*). This is in contrast with our interest: the individual behaviour and subsequent diagnosis.

## Limitations and strengths

To formulate our diagnostic criteria of NHP depression, we translated human DSM-5 criteria of depression to observable behaviours in NHPs. We operationalised the signs of depression in concrete and observable behaviours across different species. Compared to the human diagnostic criteria, our NHP criteria may be too loose: our diagnostic criteria require the presence of four out of six observable signs of depression, compared to five out of nine for humans. However, we consider the less strict diagnostic criteria justified to prevent false negative diagnoses, since three out of nine symptoms of human depression have no equivalent observable behaviours in NHPs.

In addition, although this review spans a range of primate taxa, 16 out of the 21 reviewed studies are on macaques. So, this sample does not equally represent the entire primate order. Similarly, the behavioural measures of signs of depression were mostly based on studies in macaques and apes. Therefore, some caution is advised when applying methods to taxa that they were not directly developed for.

In experimental settings, within-individual or between-group comparisons can be made to study the differences in signs of depression across conditions. However, in the reviewed cases of naturally occurring signs of depression, no pre-depression data or control groups are available for comparison. Thus, these conclusions rely on anecdotal reports or correlational evidence rather than experimental evidence. The lower level of evidence in the studies in non-experimental contexts excludes causal inferences and limits the reliability of assessments.

We aimed to diagnose depression using minimally labour intensive, minimally intrusive, and most easily accessible methods. However, for some signs of depression we could not reach these goals. For example, assessing depressed mood and cognitive deficits requires tasks.

Furthermore, although we identified several potential risk factors of signs of depression in NHPs, our review does not provide a comprehensive assessment of the prevalence or severity of each risk factor. We only included studies where signs of depression were observed, which yields an unrepresentative sample for comparing risk factors.

A strength of our review is the addition of new insights into non-human depression, adding onto other work in the field of NHP depression research. Compared to previous depression research on chimpanzees (*Ferdowsian et al., 2011*), we argue that our criteria are more concrete and only directly observable behaviours are included. Compared to a recent review on animal models of human mood (*Bliss-Moreau & Rudebeck, 2021*), our review focused on criteria for diagnosing DSM-5-derived signs of depression in NHPs through behavioural observation, rather than focussing on neurobiology and affective states. Another review identified similarities in animal models of depression and common captive animal housing (*Lecorps, Weary & von Keyserlingk, 2021*). We added to this the recognition of specific signs of depression associated with other risk factors, in both wild and captive conditions. A different review focused on depression-like behaviours in NHPs (similar to our article), along with anxiety-like behaviours (*Ausderau et al., 2023*). The latter two reviews (*Ausderau et al., 2023*; *Lecorps, Weary & von Keyserlingk, 2021*), however, were not focused on diagnosing NHP depression, but more on broadly exploring which NHP behaviours are associated with depression, and on reflecting on the ethics of researching depression-like and anxiety-like behaviours in NHPs. Our review added to this by providing clear diagnostic criteria of NHP depression and in providing a translation to observable behaviours in NHPs. Another recent review (*MacLellan et al., 2021*) showed parallels with our method, but there are differences in diagnostic criteria. For example, we added anticipatory behaviour as a measure of anhedonia and use different methods to assess cognitive deficits. Moreover, whereas this review concerned depression in poor housing conditions (*MacLellan et al., 2021*), we included a broader set of risk factors of depression in NHPs. In sum, our work added onto previous work by providing a new approach to NHP depression.

## Future research

First of all, more research into observing co-occurrence of signs of depression in individual NHPs is needed. In the reviewed literature, typically only a few signs were measured on group level. To assess depression in individual NHPs, as many signs as possible must be measured on an individual level. This may come with difficulties, as this requires ongoing baseline data collection to compare within-individuals when depression is suspected. An alternative may be assessing baseline levels of depressive parameters in healthy individuals, which also allows recognition of species-specific differences in the expression of signs of depression. In addition, age differences in depression as described in humans, such as irritability as a sign of depression in young individuals (*Sherwood et al., 2021*), might also

be discovered. Based on these data, deviations from the species baseline can be identified. Nevertheless, truly objective assessment of depression (*Rosati et al., 2012*) requires experimental studies, so this cannot be attained in principle for naturally occurring depression. If research into the co-occurrence of signs is conducted, we expect that more NHPs will be diagnosed with depression according to our criteria.

Furthermore, we encourage further development of methods measuring signs of depression. First, we recommend research to potentially identify easily observable behaviours that can serve as a proxy for depression or for signs that currently require tasks to measure. These methods should take species-specific social behaviour, signals and reactions into account. Second, we encourage research into observing the three signs of depression that are currently not (independently) observable: fatigue, feelings of worthlessness/guilt, and suicidality.

In addition, researching the effect of antidepressants on the proposed indicators of in NHPs can help improve the validity of these indicators. Although there is recent criticism on the efficacy rates of antidepressants (*Moncrieff, 2018*) and antidepressants can also be used to treat other conditions (*Bandelow et al., 2015*), exploring the effect of antidepressants may provide valuable insights. Additionally, it is important that depression assessment is done by experts who are familiar with the species and behavioural observations, because possible interventions, like prescribing antidepressants, should only be carried out when there is a strong basis to do so.

The administration of antidepressants is one intervention that can reverse signs of depression in experimentally induced depression (*e.g.*, *Harlow & Suomi, 1974*; *Qin et al., 2015*) and even in naturally occurring depression (*Chu, 2019*). Alternatively, some types of enrichment can reduce signs of depression (*MacLellan et al., 2021*). Other interventions may be to address the risk factors, if possible. If an individual expresses (signs of) depression while being housed without access to a social group (*Hennessy, Chun & Capitanio, 2017*; *Jackson et al., 2023*; *Li et al., 2013*), the opportunity for social interactions could be increased, for example in social group housing. This is in line with legislation in the European Union, where social housing of primates is obligatory (EU Directive 2010/63). Evidently, not all risk factors can be prevented, such as status loss (*Kummer, 1995*; *Setchell & Dixson, 2001*) or maternal death (*Goodall, 1986*). Thus, for some risk factors, a diagnosis of depression could aid in assessing the severity of the effect of the risk factor and allow for interventions.

Due to our inclusion criteria, our review could not provide an overview of the prevalence or severity of the risk factors. Therefore, we recommend future research specifically into risk factors of (signs of) depression in NHPs. Furthermore, although the scope of our review was limited to signs of depression attributable to one event, our diagnostic criteria are also applicable to animals subjected to an accumulation of stressors. Examples include consistent maltreatment by humans (*Lopresti-Goodman, Kameka & Dube, 2012*), insufficient housing conditions (*Lecorps, Weary & von Keyserlingk, 2021*), repeated loss (*Rasmussen & Reite, 1981*), and chronic stress (*Teng et al., 2021*; *Wang et al., 2013*; *Zhang et al., 2016*). Evidently, and beyond the scope of our review, there are non-behavioural parameters that can help identify depression. Biomarkers such as the levels of pro-inflammatory cytokins (*MacLellan et al., 2021*) or heart rate variability

(*Jarczok et al., 2018*) are associated with depressive symptoms in humans and signs of depression in NHPs. For assessing sleep changes, electroencephalograms (EEGs) are widely used in NHPs (*Li et al., 2022*; *Reinhardt, 2020*). Ideally, behavioural research into NHP depression should complement and be complemented by research into non-behavioural parameters associated with depression.

## CONCLUSION

We proposed diagnostic criteria for depression in non-human primates (NHPs). We operationalised signs of depression with a specific and limited number of observable behaviours, derived from the DSM-5 diagnostic criteria of human depression, supplemented with NHP specific behaviour. We confirmed the presence of four out of six observable signs of depression in NHPs. However, only one group of six NHPs met our criteria for a depression diagnosis. We reason that the scarcity of diagnoses can be attributed to a lack of research into co-occurrence of signs. Therefore, we call for future research into the co-occurrence of signs of depression in individual NHPs. We hope that this review and future research can contribute to researching depression in NHPs, thereby informing us about the evolutionary history and function of depression in humans, resulting in better NHP welfare and human well-being.

## ACKNOWLEDGEMENTS

We thank Annet Louwerse from the Biomedical Primate Research Centre in Rijswijk, the Netherlands for the information provided on behaviour following status loss of long-tailed and rhesus macaques.

### Funding
The authors received no funding for this work.

### Competing Interests
The authors declare that they have no competing interests.

### Author Contributions
- Jonas C. P. van Oosten conceived and designed the experiments, performed the experiments, analyzed the data, prepared figures and/or tables, authored or reviewed drafts of the article, and approved the final draft.
- Annemie Ploeger conceived and designed the experiments, performed the experiments, authored or reviewed drafts of the article, and approved the final draft.
- Elisabeth H. M. Sterck conceived and designed the experiments, authored or reviewed drafts of the article, and approved the final draft.

### Data Availability
   This is a literature review.

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
