# Peer review of "Recognising depression in non-human primates: a narrative review of reported signs of depression"

_PeerJ, doi:10.7717/peerj.18766_

## Round 0.1 · original submission · Minor Revisions

The two reviewers, who are prominent in your field of research, were very positive about your paper. The changes they request are well-considered and constructive; Taking their suggestions to heart as you revise the paper will undoubtedly make it clearer to your readers and increase its impact (and citation rate!). Please know that the slowness of this review process owes to the difficulty I had in finding persons willing to perform the review, not to the reviewers' response times nor to my inattention; that seems to be a tendency in science these days.

Please submit a point-by-point list of what you did to revise or answer the reviewers' comments. That will not only help me, but it will clarify your own thoughts about your work.

·

Basic reporting

This generally well written manuscript first tackles the question of whether nonhuman primates (NHPs) experience depression, and then proposes NHP specific diagnostic criteria for depression based on the DSM-5 definition. This is an interesting question, and this work represents a reasonably comprehensive review of the relevant literature. The breadth of interest in this review may be limited to the small community of individuals working with NHPs, but within that community multiple stakeholders (including but not limited to: biomedical researchers, behavioral researchers, field/conservation researchers, zoo/sanctuary staff, captive primate care staff) should find this work interesting and useful. There are at least two other recent reviews similar to this paper though this paper does provide a potential new paradigm for evaluating the already available information. The introduction makes a clear case for why it is important to be able to recognize if an NHP is depressed.

Experimental design

• The search criteria included the term nonhuman primates but not monkey or ape or any of the other appropriate terms. If monkey is used the initial search results in far more papers. This suggests that many potentially important papers were missed in this review.
• Information, ideally in the form of a flow chart or table, should be included to show the number of papers found by the database searches, number found by further review of cited articles, number of papers that were excluded and for what reasons, and the number of papers that were included in the final assessment.
• This paper purports to evaluate information available in the literature on depression in NHPs. In reality, with the exception of one report in chimpanzees, all of the information is from cercopithecines, with the majority of the information from the single genus Macaca. This is not a comprehensive assessment of NHPs. Consider adjusting the title and language throughout to clarify the limited number of NHPs species that are reviewed.
• There are significant species differences that could impact behaviors included in the proposed criteria. If this is a general NHP definition, there needs to be a way to account for these species-specific differences.
• This manuscript would be strengthened if the authors suggested ways that could be explored to test components of the DSM-5 that they deem currently untestable in NHP. That would move this paper beyond a review that just categorizes what exists to something that has the potential to move the field forward.
• Justification for the number of criteria and the 2-week timeframe need to be strengthened.
• In the human definition of depression, symptoms must occur within the same 2-week period. In the proposed NHP definition, symptoms must be continuous throughout the 2-week period. This seems to be a much stricter condition.
• The justification for only including studies that were attributable to one change in environment versus arrays of multiple stressors should be strengthened and possibly reconsidered. It is reasonable that depression can result from the accumulation of stressors.

Validity of the findings

• The natural social structure of the species must be considered in interpreting the results from the different tasks. This does not appear to be considered in your evaluation.
• Different methods are discussed and then dismissed because they haven’t been used previously in the specific context of interest. Perhaps it would be useful if all of these potential methods were included with information about what validations need to be performed and what considerations are important for each method.
• Line 302: What counts as a significant weight change. If weight is assessed solely by visual observation, there need to be caveats for body size vs. adiposity and coat quality at a minimum.
• Lines 348-350 and 369-370. Elaborate on why open field tests are not a valid indicator of psychomotor agitation or retardation.
• Some behaviors, e.g. hunching as an indicator of psychomotor retardation, are deemed acceptable indicators but are not reasonable for all of the species of NHP. These species-specific limitations should be clearly described.
• Measurement of fatigue. Available measurements (that are used in humans and rodents) are discounted because they are already accepted by your paradigm for assessments of other DSM components. This is reasonable but perhaps fatigue can be represented differently? Consider a metric that combines activity and sleep levels.
• They acknowledge that assessing sleep patterns can be very labor intensive, but don’t discuss important potential difficulties such as how and where animals sleep.
• In humans, depression can look different in different age groups. Age differences need to be discussed and considered in your definition.
• For much of the literature reviewed the signs of depression could not meet the 2-week criteria due to study design issues (didn’t evaluate for that long), or they didn’t meet the # criterion required because more criteria were not evaluated. Not finding this as depression is different from it not actually being depression. This limitation is mentioned briefly (lines 663-664), but this is a major issue with this evaluation and needs greater consideration.
• For future plans a lot of space is allotted to talking about how to alleviate NHP depression but based on your findings there is still no good definition of depression so deciding when and how to treat as well as treatment efficacy would not be possible at this point. This limitation should be acknowledged.

Additional comments

• Line 302: “significant weight change can be assessed and reliably by observations,…” Is something missing from this?
• Minor: line 353: change “This can be divided in” to “This can be divided into spontaneous…”
• Minor: line 491: Change “In next part of this review…” to “In the next part of this review…”
• Minor: line 629. Should be “In other…” not “In another…”
• Minor: line 902. Should be “in NHPs.” not “to in NHPs.”.

Reviewer 2 ·

Basic reporting

The field of depression in animals has been reviewed recently but the current review provides a different assessment of behavioral criteria for depression in animals while reviewing all published studies, The authors also provide an excellent discussion of the novelty within their review compared to those published.

Experimental design

No additional comments

Validity of the findings

No comment

Additional comments

The authors provide a review of whether NHPs can experience depression by comparing reported behavioral signs of depression in NHPs in the literature with diagnostic criteria of human depression based on the DSM-5. In this review, the assessment of depressive signs in NHPs was also supplemented with NHP specific behavior. The goal was also to develop the use of observational methods over trained tasks to assess signs of depression, avoiding many difficulties with administering behavioral tasks to depressed animals. The review of published reports in studies of depression in NHPs revealed that certain criteria for depression included in the DSM-5 can be observed or measured with tasks in NHPs, but no studies reported the strict criteria of depression proposed by the authors. This paper makes a thorough review of the literature on studies of non-human signs of depression. It also provides constructive remarks on the need to complement animal studies with the use of more rigorous criteria similar to those used for humans and the addition of investigations at a more individual level. Overall, the review will likely reach an audience of researchers interested in depression and welfare in nonhuman primates.
Few points for consideration:
1- The description of the preferential looking task provided on page 9, second paragraph is missing some important information about the task for an unfamiliar reader.
a. For example, the animal response is not provided after the presentation of neutral and negative faces on the screen. The authors could add that “the looking preference of the subject for one of the expressions is recorded”.
b. Second, the authors then mentioned that a negative stimulus was presented before and after the task; the use of “negative stimulus” is here ambiguous because are we talking about another negative expressions or something else?…The reader needs to wait the end of the paragraph to learn that the negative stimulus was not an expressive face but “a stressful veterinary check”. This info needs to be placed earlier in the sentence when the term “negative stimulus” is used.
2- On page 10, first paragraph, the authors noted: “Notably, the dot-probe task has not yet been used to study the effect of negative affect on attentional biases in NHPs”. The dot-probe task was already described for use with rhesus monkeys (Macaca mulatta) and was shown to be useful to measure affective attentional bias. See Parr L et al, Psychoneuroendocrimology, 2013, 38:1748.
3- On Page 15, end of second paragraph: Please remove “is” in the sentence: “…which is can be stressful for animals”.
4- On page 15: after reviewing the procedures to measure circadian sleep patterns, the authors concluded: ”In sum, for assessing sleep disturbances in NHPs, observing the full diurnal cycle of an animal may be the most valid, but hardly feasible”. I disagree with this statement because the use of accelerometers to measure activity in captive and wild NHPs is very easy since these accelerometers are placed in collars fitted on the animals’ neck and the collars could be placed while the animals are sedated for regular health investigation. There exist even more recent procedures to download data stored in the accelerometters remotely without disturbing the animals and these procedures could provide an easy and accurate estimate of sleep patterns.
5- On page 20: the authors described the recognition task, which is in fact a preferential viewing task using neutral stimulus, such as objects, as opposed to expressive faces as described earlier in their review. What is important to notice here is that the task used different time intervals between the familiarization and test to measure recognition memory. This task has been extensively used now to assess recognition memory processes and their neural substrate in both humans, NHPs, and rodents of different ages, including infants, juveniles, adolescents and adults (see Ennaceur, A and De Souza Silva, MA Handbook of Object Novelty Recognition, Academic Press: London, 2018). It could be easily used to measure memory processes in depressed animals.

---

## Round 0.2 · Minor Revisions

The letter above is a "canned" doc that I can't figure out how to modify; please attend only to this paragraph. Your revisions and responses to the reviewers' comments are good, so I see no reason to send the Ms out again. Well done.
Please check the entire Ms for minor errors, such as punctuation and a couple of split infinitives. Reading the paper aloud to co-authors or an assistant is an effective method. The paper contains excess verbiage, and should be shortened. I'm hoping for a reduction of 20% or so. For example, omit unnecessary adverbs, and eliminate phrases such as "we conclude that" and "they found that". Why not ask a colleague to help with reducing the length of the paper? These further changes may be annoying to deal with now, but paper will be improved and will have greater impact when you're done.

---

## Round 0.3 · accepted · Accept

Well done! Thank you for attending to the various details through these 2 revisions. It was a pleasure to edit your paper.